# Effects of Surface Modification with Stearic Acid on the Dispersion of Some Inorganic Fillers in PE Matrix

**Thanh Tung Nguyen** *, **Van Khoi Nguyen, Thi Thu Ha Pham, Thu Trang Pham** * and **Trung Duc Nguyen**

Institute of Chemistry, VAST, 18 Hoang Quoc Viet, Cau Giay, Ha Noi 122300, Vietnam;
khoinguyen56@gmail.com (V.K.N.); haptt6@gmail.com (T.T.H.P.); ducnt224@gmail.com (T.D.N.)
* Correspondence: nttung@ich.vast.vn (T.T.N.); thutrang90vhh@gmail.com (T.T.P.); Tel.: +84-90-133-3885 (T.T.N.);
+84-38-997-0517 (T.T.P.)

**Abstract:** To evaluate the effects of surface modification with stearic acid on the dispersion of some inorganic fillers in polyethylene (PE) matrix, masterbatches containing 20–40 wt% of stearic acid uncoated and coated inorganic fillers and the linear low-density polyethylene (LLDPE) films containing 3–7% stearic acid uncoated and coated inorganic fillers were prepared. Two types of inorganic fillers used in the masterbatch included bentonite and silica. The structural change of inorganic fillers, whose surface was modified with stearic acid, was studied using IR spectroscopy. The dispersion of inorganic fillers in LLDPE matrix was evaluated using scanning electron microscope (masterbatch samples) and optical microscope (film samples). Changes in the melting temperature of LLDPE in the presence of inorganic fillers were evaluated by using differential scanning calorimeter (DSC). The mechanical properties of the films were evaluated according to ASTM D882. Surface-treated fillers with stearic acid dispersed in the masterbatches and films better than untreated fillers did. Stearic acid did not change the melting temperature of the filler/PE masterbatches. The mechanical properties of the films containing stearic acid coated fillers were higher than those containing unmodified fillers.

**Keywords:** polyethylene; bentonite; silica; inorganic fillers; stearic acid

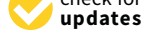



## 1. Introduction

Polyethylene (PE) is the most commonly used material in the packaging industries due to its many advantages, such as being inert, inexpensive, readily available, easy to process, flexible, and semi-permeable [1]. To change the thermal, mechanical, and permeability properties of polymer, and to reduce the production costs, inorganic fillers, such as metal oxide, metal powders, carbon black, silica, spherical or fibrous glass [2], talc, calcite [3,4], kaolin, mica [5], etc. are added. At present, modified atmosphere packaging is increasingly receiving research attention; it is widely used to prolong the shelf life of fresh fruits and vegetables. Inorganic fillers, which are incorporated in the plastic films to regulate gas and water vapor permeability of film, are usually zeolite, bentonite, and silica [6–9]. However, due to the hydrophilic nature of these additives, it is necessary to treat their chemical surfaces before mixing them to increase their compatibility and dispersion into the PE matrix. To create the hydrophobic surface with the aim of increasing the compatibility of fillers with polymer molecules, fatty acids, such as oleic acid, stearic, palm oil [10–12] are often used. Stearic acid is most commonly used because of its advantages such as low cost and easy processing. In addition, when modifying the filler surface with stearic acid, it reduces water absorption, prevents the agglomeration of the filler particles, and reduces the surface energy to help them disperse easily into the PE matrix.

Hyun K. et al. reported that the zeolite surface modification with stearic acid had improved the flexibility of the low-density polyethylene (LDPE), linear low-density polyethylene (LLDPE), high-density polyethylene (HDPE) matrices. Stearic acid modified zeolite dispersed in composites more evenly than unmodified zeolite [13]. Zaharri N. et al. also

reported that the mechanical properties of zeolite/propylene composite in the presence of stearic acid coupling agent was better than without stearic acid [14]. Angle et al. found that the elongation at break of the composites containing modified bentonite increased, and the dispersion of bentonite was better in comparison with those containing unmodified bentonite [11]. Therefore, the aim of this study is to evaluate the dispersibility of stearic acid modified and unmodified inorganic fillers (bentonite, silica) in PE matrix.

## 2. Materials and Methods

### 2.1. Materials

Linear low-density polyethylene (LLDPE) plastic resin produced by Formosa Plastics Corporation (Yunlin, Taiwan) was used as the matrix material. Its density was 0.924 g/cm$^3$, and its melt flow index (MFI) was 21 g/10 min at 2.16 kg/190 °C.

Bentonite (Be) with an average particle size of 18.52 μm was supplied by Minh Ha Co., Ltd. (Bac Giang, Vietnam). Silica 606 (Si) with an average particle size of 15.86 μm was supplied by Gia Tuong Co., Ltd. (Gia Lai, Vietnam). Stearic acid was used directly without refining.

### 2.2. Inorganic Fillers Surface Treatment

To prepare the stearic acid coated inorganic fillers, 17.04 g of stearic acid was added to a 1-L beaker, and then 400 mL of ethanol and 200 mL of distilled water were added. The mixture was heated to 70 °C and stirred constantly until the stearic acid was completely dissolved. Next, 100 g of inorganic fillers (bentonite, silica) were added to stearic acid solution, stirred constantly at 70 °C for 5 h. Then the mixture was left to settle, and finally be filtered to get the product. Inorganic fillers coated with stearic acid (Be/St, Si/St) were dried in a vacuum oven for 6 h at 60 °C, then ground to form powder [13].

### 2.3. Masterbatch Preparation

To achieve a good dispersion of inorganic fillers in plastic films, the fillers were first mixed with the resin in the masterbatch form. Inorganic fillers (bentonite, silica) and LLDPE resin were dried at 80 °C for 8 h. These additives and LLDPE resin were initially melt, and then mixed in an internal mixer model Plastograph$^®$ EC (Duisburg, Germany) at 160 °C with a rotor speed of 50 rpm for 7 min. The masterbatches of LLDPE and fillers were compressed and molded into 1-mm-thick plates on a Gotech hot press at 170 °C at a pressure of 50 kg/cm$^2$ for 5 min. To evaluate the dispersion of the fillers on the PE matrix, the fillers were added into PE with different contents (0%, 20%, 25%, 30%, 35%, and 40%).

### 2.4. Film Preparation

LLDPE film samples were fabricated with a thickness of 30 μm by extrusion blowing using an XD-35 extruder with the screw diameter of 35 mm and the L/D ratio of 28:1. The surface activated/inactivated fillers with stearic acid were incorporated into the film formulation with a concentration of 5% by using filler/PE masterbatch, which loaded 30% of coated fillers. The film sample symbols were PE-5Be, PE-5Be/St, PE-5Si, and PE-5Si/St, respectively.

### 2.5. Characterization

2.5.1. Mechanical Properties

Tensile properties of film specimens were measured by Instron 5980 Testing Machine (Illinois Tool Works, Inc., Norwood, MA, USA), according to ASTM D882 at a crosshead speed of 10 mm/min.

2.5.2. Scanning Electron Microscope (SEM)

The fracture surface morphology of masterbatches and the surface morphology of films were studied by a scanning electron microscopy (SEM) (JEOL 6490, Tokyo, Japan) at 15 kV. The surface of the samples was coated with platinum using a sputter coater before examination.

### 2.5.3. Fourier Transfer Infrared Spectra (FTIR)

A Fourier transform infra-red (FTIR) spectrometer (Nicolet Impact model 410, Nicolet, Madison, WI, USA) was used to observe the chemical changes in structure of inorganic fillers before and after treating their surface. The equipment was operated with a resolution of 4 cm$^{-1}$, and scanning ranged from 4000 to 500 cm$^{-1}$. IR spectra of liquid and solid samples measured by using a KBr disk and pelletizing with KBr, respectively.

### 2.5.4. Differential Scanning Calorimetry (DSC)

The melting behavior of the film samples was studied by a NETZSCH DSC 204F1 Phoenix differential scanning calorimetry (Netzsch, Selb, Germany). The samples were first heated at a rate of 10 °C/min from rt to 200 °C, followed by being cooled down to rt at the same rate.

## 3. Results and Discussions

### 3.1. Fourier Transfer Infrared Spectra of Unmodified and Modified Fillers

The FTIR spectra of stearic acid and unmodified and modified inorganic fillers are presented in Figure 1.

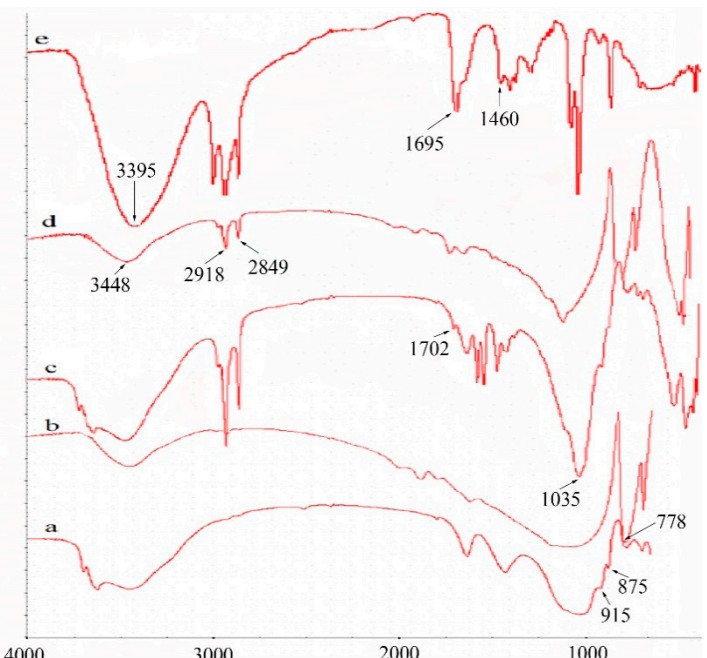

**Figure 1.** The FTIR spectra of (**a**) unmodified bentonite, (**b**) unmodified silica, (**c**) modified bentonite, (**d**) modified silica, (**e**) stearic acid.

Regarding IR spectrum of stearic acid: the strength peak at 3395.4 cm$^{-1}$ was assigned to –COOH functional group, the strength peak at 1695.7 cm$^{-1}$ was assigned to the stretching of C=O, the peak at 1460.4 cm$^{-1}$ was assigned to the scissoring of -CH$_2$, the peak at 1297.7 cm$^{-1}$ was assigned to the stretching of C-O [11], and the peak at 2916.5 cm$^{-1}$ and 849.0 cm$^{-1}$ was assigned to the asymmetric and symmetric stretching of CH$_2$, respectively [15].

Comparing the IR spectra of unmodified and modified inorganic fillers with stearic acid showed that the IR spectra of stearic acid coated inorganic fillers (bentonite, silica) showed some new peaks, which were near 2920 and 2850 cm$^{-1}$, corresponding to stretching of CH$_2$, near 1700 cm$^{-1}$, corresponding to the stretching of C=O. Particularly, the IR spectrum of modified bentonite showed additional peaks in 1400–1480 cm$^{-1}$ and 1500–1600 cm$^{-1}$, which was assigned to the symmetry flex oscillation and the dissymmetry flex oscillation of COO [16]. This indicated that a chemical reaction between stearic acid

and bentonite, silica had occurred. Li et al. [17] obtained similar results when modifying wollastonite with stearic acid. Meng and Duo [16] also reported a chemical reaction between wollastonite and pimelic acid. In contrast, in the investigation of Angel et al. [11] and Gonzalez et al. [18], the interactions between bentonite or montmorillonite and stearic acid were mainly physical not chemical. In addition, in the spectra of unmodified and modified fillers, the strength peak at about 1000 cm$^{-1}$ was assigned to the stretching vibrations of Si-O-Si, the peak at about 780 cm$^{-1}$ was assigned to the symmetric vibration of Si-O. In the spectra of unmodified and modified bentonite, the peak at 915 and 875 cm$^{-1}$ was assigned to the bending vibration of AlOH and AlMgOH, respectively.

### 3.2. Dispersion of Filler Particles in Masterbatches

Morphology of the fracture surface was an effective method to evaluate the dispersion of filler particles in PE matrix. The SEM micrographs of fracture surfaces of untreated as well as treated filler/PE masterbatches are shown in Figures 2 and 3.

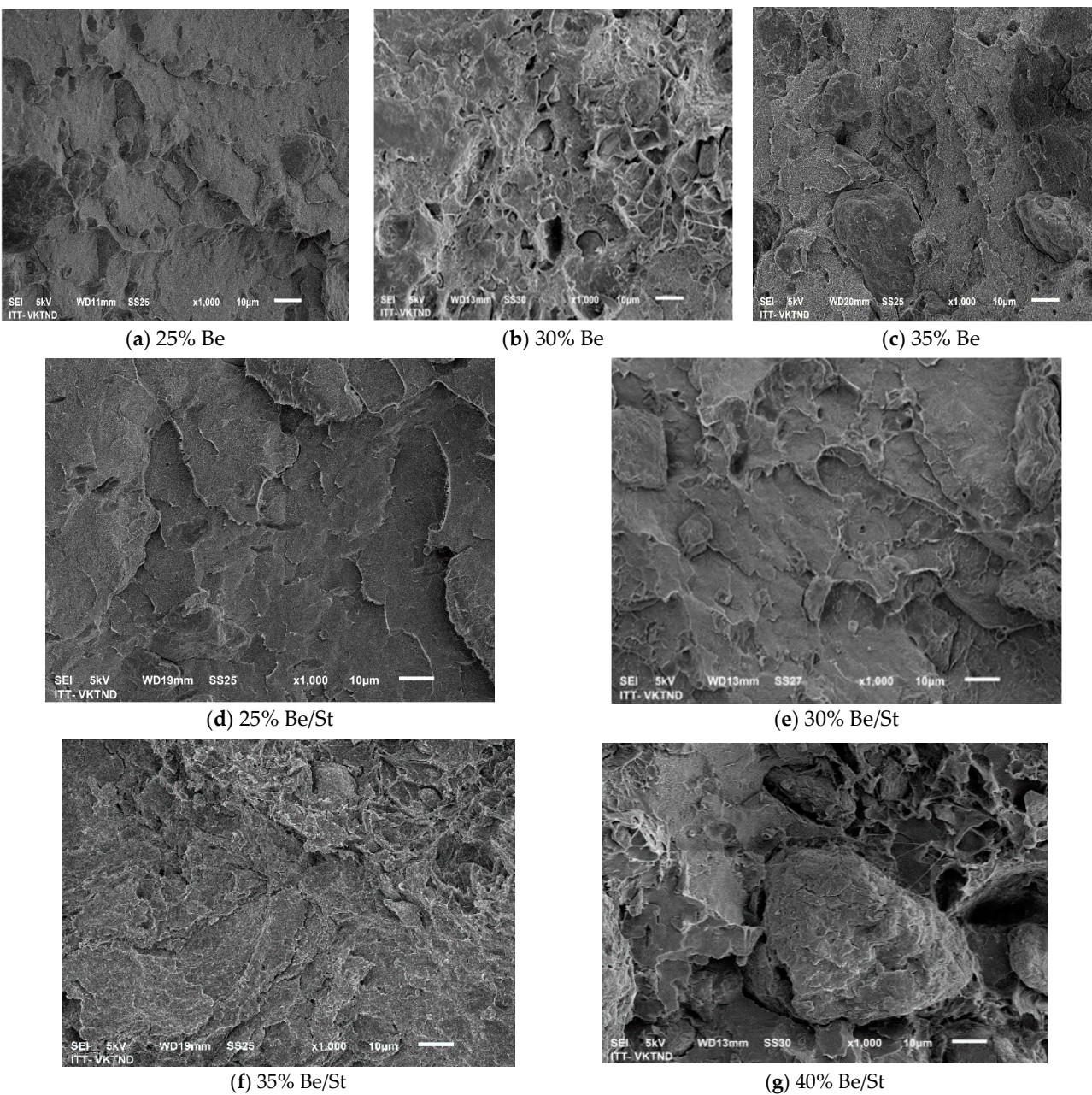

**Figure 2.** The morphology of the fracture surface of untreated and treated bentonite/PE masterbatches.

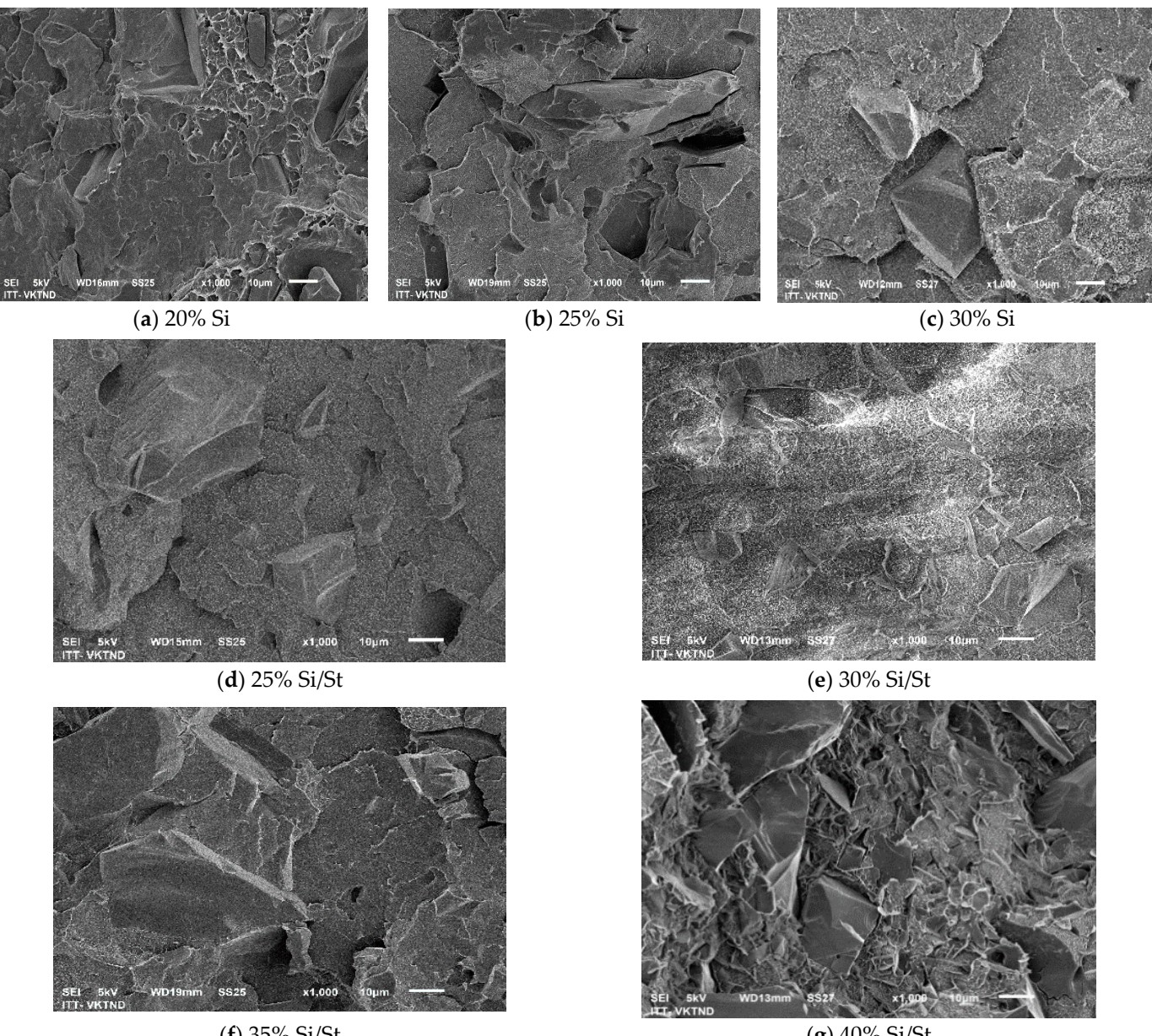

(**a**) 20% Si  (**b**) 25% Si  (**c**) 30% Si

(**d**) 25% Si/St  (**e**) 30% Si/St

(**f**) 35% Si/St  (**g**) 40% Si/St

**Figure 3.** The morphology of the fracture surface of untreated and treated silica/PE masterbatches.

It could be seen that the population density of the inorganic fillers increased as the inorganic fillers content increased. The results showed that in samples containing uncoated inorganic fillers, the inorganic fillers particles dispersed still relatively uniformly at the bentonite content of 25% and the silica content of 20%. However, when increasing the uncoated bentonite content to 30–40% and the uncoated silica content to 25–40%, the inorganic filler particles tended to form agglomeration, not even dispersed in the PE matrix. This was explained as the inorganic filler particles had high surface energy; therefore, distances between particles were decreased when the density of particles was increased, so there was a possibility of surface interaction among the particles leading to agglomeration.

In samples containing stearic acid coated inorganic fillers, it was found that in the studied concentration range (20–35%), the inorganic filler particles still uniformly dispersed in the PE resin. However, both stearic acid coated bentonite and modified silica had poorer adhesion to the PE matrix at 40% content. This could be explained that the surface treatment with stearic acid not only significantly reduced the free energy of the filler particles surface but also the interaction between filler particles, leading to a better dispersion of the filler

particles in the PE matrix and a reduction in agglomeration of particles [11]. Similar to zeolites, bentonite and silica also interacted with stearic acid through silanol groups [19,20]. Luis et al. proposed two interaction modes of fatty acid on the zeolite surface: (1) adsorption of –COOH group of fatty acid by H-bonding to Al-O-Si groups of zeolite and (2) physical adsorption by London forces of -CH$_3$ groups of fatty acid and Si-O-Si groups of zeolite [21].

### 3.3. Differential Scanning Calorimetry of Masterbatches

The thermal properties of the masterbatches with different filler contents were investigated by differential scanning calorimetry (DSC). The DSC results for fillers/PE masterbatches are shown in Figures 4 and 5 and summarized in Table 1.

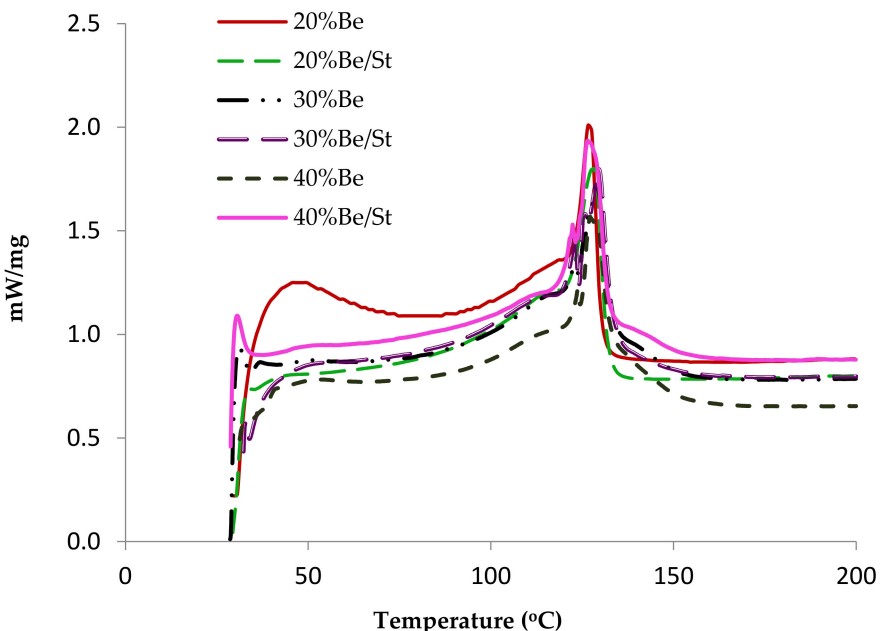

**Figure 4.** The DSC curve of the bentonite/PE masterbatches.

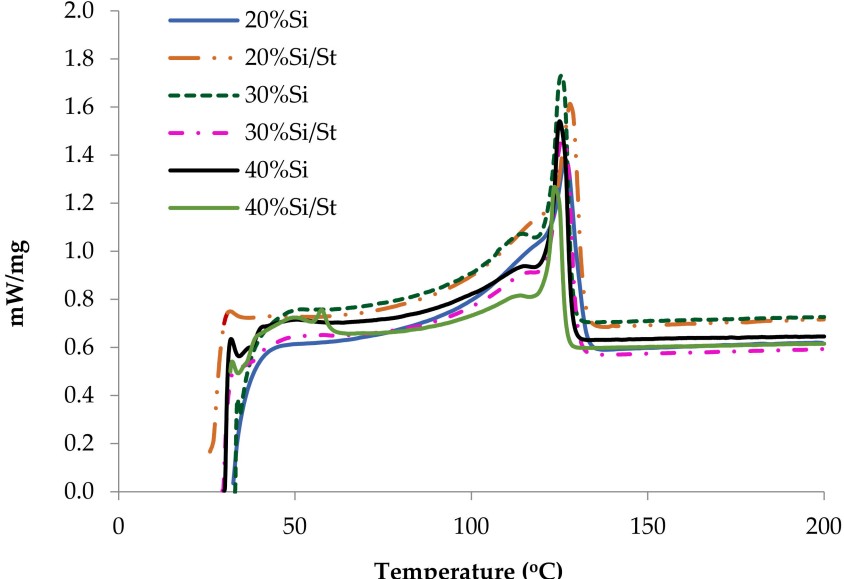

**Figure 5.** The DSC curve of the silica/PE masterbatches.

**Table 1.** The melting temperature of the masterbatches.

| Filler Content | Melting Temperature of Samples (°C) | | | |
| --- | --- | --- | --- | --- |
| | Uncoated Bentonite | Coated Bentonite | Uncoated Silica | Coated Silica |
| 0 | 127.8 | | | |
| 20% | 127.0 | 128.0 | 126.7 | 128.1 |
| 30% | 129.0 | 129.6 | 125.5 | 125.8 |
| 40% | 127.1 | 126.5 | 125.0 | 125.3 |

The results showed that there was no difference in the melting temperature between the filler/PE samples and the stearic acid coated filler/PE samples. The results also showed that the melting temperature of the masterbatches containing both uncoated and coated fillers (bentonite and silica) was not significantly different when compared with pure PE, which indicated that adding bentonite and silica did not change the characteristics of polymeric materials. Daiane et al. also obtained similar results when incorporating silver-exchange zeolite Y into polyethylene [22]. Pehlivan et al. also found that incorporating zeolite into polypropylene did not change the melting temperature of polymer [23]. In contrast, Rund et al. found that bentonite and diatomite reduced the melting temperature of PE/cellulose composite [8].

*3.4. Optical Micrographs of the PE Films*

Figure 6 showed optical micrographs of the inorganic fillers filled LLDPE films with a filler concentration of 5%.

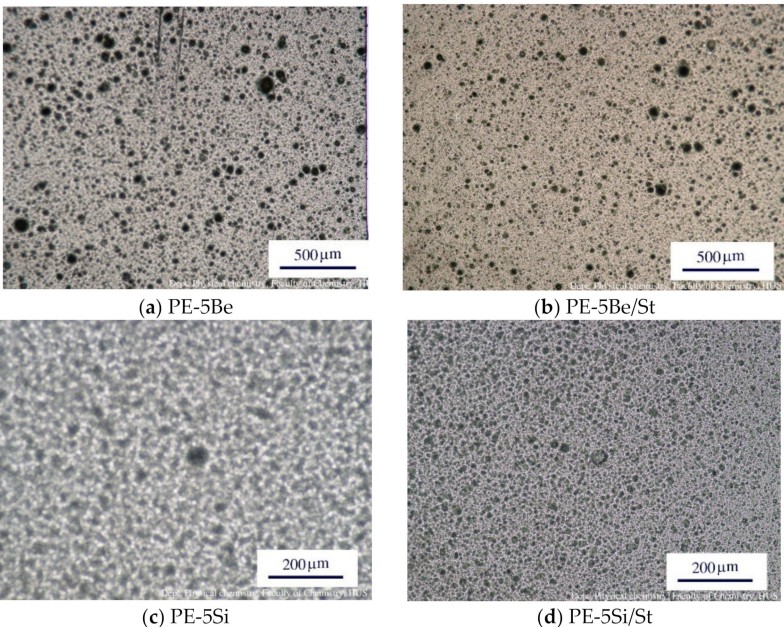

(**a**) PE-5Be

(**b**) PE-5Be/St

(**c**) PE-5Si

(**d**) PE-5Si/St

**Figure 6.** Optical micrographs of bentonite and silica filled LLDPE films.

The particle–particle interactions led to the agglomeration and heterogeneous distribution of filler particles. In the film samples using unmodified bentonite and silica, the agglomeration of filler particles was observed. By comparing film samples with the same filler content, it could be seen that the stearic acid coated filler particles dispersed uniformly with smaller particle sizes into the PE matrix. Thus, stearic acid acted as a binder to increase the adhesion between the filler particles and the PE matrix, reducing the agglomeration of filler particles. Angel et al. [11] also found that adding fatty acids as an additive to PP film formulations improved the polymer–filler interactions due to changing surface interactions.

*3.5. The Mechanical Properties of the Films*

The mechanical properties of film were of major importance for all applications. Table 2 showed the results of the mechanical properties of pure PE film and inorganic fillers filled polyethylene films.

**Table 2.** Mechanical properties of the films.

| Sample | Tensile Strength (MPa) | Elongation at Break (%) |
|---|---|---|
| PE | 26.98 ± 0.3 | 776.50 ± 4.23 |
| PE-5Be | 18.25 ± 2.43 | 308.10 ± 6.34 |
| PE-5Be/St | 21.67 ± 2.93 | 431.23 ± 22.01 |
| PE-5Si | 18.57 ± 0.71 | 359.94 ± 21.22 |
| PE-5Si/St | 23.52 ± 1.81 | 476.19 ± 28.90 |

It could be seen that the mechanical properties of pure PE film were higher than that of uncoated and coated inorganic fillers filled polyethylene films. This was because fillers particles limited the mobility of the polymer molecules and formed stress concentration points [14].

The results showed that both the tensile strength and elongation at break of the PE films containing both uncoated bentonite and silica decreased sharply. The tensile strength of PE-5Be and PE-5Si films decreased to 8.25 MPa and 8.57 MPa, whereas elongation at break of PE-5Be and PE-5Si films decreased to 108.10% and 159.94%, respectively. When treating the fillers with stearic acid, the mechanical properties of the PE films containing coated fillers significantly improved, which were much higher than those of PE films containing uncoated filler. This finding was also supported by the results of the optical microscope mentioned earlier. When bentonite and silica were treated with stearic acid, they distributed more uniformly in the PE matrix with a smaller size. According to Angel et al., in addition to acting as an interface modifier, stearic acid also acted as a lubricant, leading to a decrease in the interaction between polymer molecules, thereby increasing the elongation at break [11].

## 4. Conclusions

The effects of surface modification of bentonite and silica with stearic acid on thermal properties, filler dispersion of filler/PE masterbatches, mechanical properties, and filler dispersion of filler/PE films were investigated. IR spectroscopy showed that the chemical interaction between bentonite and stearic acid occurred. The melting temperature of PE masterbatch containing modified fillers did not change much compared to that of PE masterbatch containing unmodified fillers. The bentonite did not change the melting temperature of the PE matrix, but the silica reduced its melting temperature. The PE films containing modified inorganic fillers had higher mechanical properties than those containing unmodified inorganic fillers. At the same time, the surface modified inorganic fillers by stearic acid dispersed in the masterbatches and films better than unmodified inorganic fillers did, and since stearic acid reduced the agglomeration of filler particles, it could be incorporated into the masterbatch as well as the film with a higher concentration.

**Author Contributions:** Conceptualization: T.T.N. and T.T.H.P.; data curation: T.T.P., T.D.N.; formal analysis: T.T.N. and T.T.P.; investigation: T.T.N. and T.T.P., T.D.N.; methodology: T.T.N. and T.T.H.P., T.D.N.; project administration: T.T.N.; software: T.T.H.P. and T.T.P.; writing—original draft: T.T.H.P. and T.T.P.; writing—review and editing: V.K.N. and T.T.P. All authors have read and agreed to the published version of the manuscript.

**Funding:** This research received no external funding.

**Institutional Review Board Statement:** Not applicable.

**Informed Consent Statement:** Not applicable.

**Conflicts of Interest:** The authors declare no conflict of interest.

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
