# Peer review of "Effects of Surface Modification with Stearic Acid on the Dispersion of Some Inorganic Fillers in PE Matrix"

_jcs, doi:10.3390/jcs5100270_

Round 1
Reviewer 1 Report
In this paper the novelty is not enough: it is the modification of another inorganic fillers with stearic acid. My other comments are as follows:
- Figur 1: the descriptions of the axis are not visible.
- All signals visible on FTIR spectra should be described not only for stearic acid and some for modified silica.
- More details should be given as regards FTIR experiments: were the samples as KBr tablets or ATR or other?
- It is difficult to interpretate SEM images as regards dispersion of modified and unmodified fillers.
- The Authors should have agreement to use Figure 4.
- Table 1.: the error should be given. THe thermal properties are acctually unchanged.
Author Response
Dear Reviewer,
First of all, we would like to thank you very much for your comments on our manuscript.
We are very sorry for sending the manuscript back late.
We fixed the manuscript as you suggested:
- Clarified the axis of the FTIR spectra
- Described in detail the FTIR experiment for each sample type
- Described other peaks in the spectra of unmodified and modified bentonite and silica
- Enlarged SEM images of samples for easy tracking
- Fixed thermal properties as you suggested
We look forward to receiving further your comments to improve our manuscript.
Best regards,
Reviewer 2 Report
The paper presents the effects of surface modification with stearic acid on the dispersion of some inorganic fillers in PE matrix. According to the reviewer’s opinion, the paper is well-structured and clear. The topic is interesting and falls within the aim of the journal. In addition, the results are well-presented and could be helpful to further develop the same topic. Therefore, the paper can be accepted for publication in the current form.
Author Response
Dear Reviewer,
We would like to thank you very much for your comments on our manuscript.
We look forward to receiving further your comments to improve our manuscript.
Best regards,

Round 2
Reviewer 1 Report
It can be published in the present form.